# Litter Decomposition in Pacific Northwest Prairies Depends on Fire, with Differential Responses of Saprotrophic and Pyrophilous Fungi

**DOI:** 10.3390/microorganisms13081834

**Published:** 2025-08-06

**Authors:** Haley M. Burrill, Ellen B. Ralston, Heather A. Dawson, Bitty A. Roy

**Affiliations:** 1The Institute of Ecology and Evolution, University of Oregon, Onyx Bridge, 272, 1318 Franklin Blvd, Eugene, OR 97403, USA; ellen.ralston1@gmail.com (E.B.R.); dawson.heathera@gmail.com (H.A.D.); bit@uoregon.edu (B.A.R.); 2Kansas Biological Survey, University of Kansas, 2101 Constant Ave., Takeru Higuchi Hall, Lawrence, KS 66047, USA

**Keywords:** decomposition, fire, fungi, grasslands, litter bags, prairie, pyrophilous, saprotrophic

## Abstract

Fungi contribute to ecosystem function through nutrient cycling and decomposition but may be affected by major disturbances such as fire. Some ecosystems are fire-adapted, such as prairies which require cyclical burning to mitigate woody plant encroachment and reduce litter. While fire suppresses fire-sensitive fungi, pyrophilous fungi may continue providing ecosystem functions. Using litter bags, we measured the litter decomposition at three prairies with unburned and burned sections, and we used Illumina sequencing to examine litter communities. We hypothesized that (H1) decomposition would be higher at unburned sites than burned, (H2) increased decomposition at unburned sites would be correlated with higher overall saprotroph diversity, with a lower diversity in autoclaved samples, and (H3) pyrophilous fungal diversity would be higher at burned sites and overall higher in autoclaved samples. H1 was not supported; decomposition was unaffected by burn treatments. H2 and H3 were somewhat supported; saprotroph diversity was lowest in autoclaved litter at burned sites, but pyrophilous fungal diversity was the highest. Pyrophilous fungal diversity significantly contributed to litter decomposition rates, while saprotroph diversity did not. Our findings indicate that fire-adapted prairies host a suite of pyrophilous saprotrophic fungi, and that these fungi play a primary role in litter decomposition post-fire when other fire-sensitive fungal saprotrophs are less abundant.

## 1. Introduction

Fungi play a paramount role in ecosystem function, since they include plant mutualists, plant pathogens, and decomposers. Furthermore, fungal mycelia knit together soil components, contributing substantially to soil structure [1,2]. Soil nutrient cycling via decomposition is a critical function for saprotrophic fungi, which excrete extracellular enzymes that break down complex compounds such as lignin into accessible nutrients [3,4]. Fire changes the organic inputs (e.g., charcoal is not the same as wood) and the saprotrophic communities. The relative abundance of fire-adapted “pyrophilous” fungi increases post-fire, as these organisms can tolerate fire or take advantage of the high disturbance of fire, and some depend on fire-altered substrates (PyOM) [5,6,7]. Thus, pyrophilous fungi remain in the ecosystem and provide ecosystem services when other fungi are unable to survive. Litter decomposition is a combination of abiotic and biotic interactions, including saprotrophic communities, seasonal abiotic shifts, and seasonal variation in plant growth and litter inputs [8,9,10]. An environmentally driven timing of decomposition seems particularly likely in the PNW due to its Mediterranean climate. Grass biomass peaks in the spring (May), after which the grasses become dormant and brown during the summer (June–September) as there is little to no rain, and growth resumes in September with the first rains [11]. Decomposition by saprotrophic fungi may be highest under wetter but still warm fall conditions [12,13]. For these reasons, we expected litter decomposition to be highest in the fall months in unburned grasslands, as the organic matter input from senesced plants is high, temperatures are still warm, and the fall rains are beginning to saturate the soil.

Fire changes both the abiotic and biotic environments. Burned grasslands have less organic matter (both less total organic carbon [TOC], but also less woody plants) to decompose, thus decomposition rates may be lower than in unburned grasslands [14]. Given the decrease in available organic matter in burned ecosystems, and a typically positive relationship between fungal diversity and TOC [15], saprotrophic fungal diversity is likely lower immediately following a fire than in unburned areas. In addition, unburned systems retain the fungi that are not fire-adapted, so we might expect overall fungal diversity/richness to be higher in unburned grasslands. If higher diversity is correlated with higher decomposition rates as some work has shown (e.g., [16,17]), then we should see lower decomposition in burned prairies. Overall, the relative effects of fire on fungal communities and thus the fungal effects on decomposition remain under-explored.

To better understand the effects of soil microorganisms in decomposition, studies have used various sterilization methods. However, each sterilization method comes at a cost [18,19]. For example, autoclaving plant litter may kill most microorganisms, but it may also simultaneously break down complex chemical compounds in the litter that are then more accessible to the microorganisms [18]. In this case, we might expect autoclaved litter to decompose more rapidly—especially in burned grasslands, where such changes to the structure and chemistry of the litter could be favored by pyrophilous fungi. However, due to the removal of many microbial organisms (including saprotrophs) during autoclave processing, we expected autoclaved litter to decompose more slowly, as saprotrophic microbes would need to be recruited from the surrounding soils.

Fire-adapted fungi are likely the main drivers of biological decomposition, soil structure, and nutrient turnover in burned areas [2,5,20,21]. Certain Ascomycete genera like *Aspergillus* and *Phoma* degrade plant biomass faster than others [22], and decomposition by ascomycetes has been found to decrease over time, slowly being replaced by basidiomycetes [23]. Given the remaining limited understanding of pyrophilous fungi in general, the relative contribution of pyrophilous fungi to decomposition remains unexplored. We designed a study to answer the following questions: (1) Is litter decomposition affected by controlled burning? (2) Does fungal diversity depend on burning, and if so, does fungal diversity contribute to decomposition? (3) How does fire alter the saprotrophic and pyrophilous fungal community composition in prairies? Our hypotheses are shown in Figure 1.

**Hypothesis** **1.**
*Decomposition rate is overall lower in burned treatments, with untreated litter having higher decomposition compared with autoclaved litter.*


**Hypothesis** **2.**
*Fungal saprotroph diversity will be lower in burned treatments, and pyrophilous fungal diversity will be highest in autoclaved litter compared with untreated litter and will be overall higher in the burned treatment.*


If hypotheses 1 and 2 are supported, then:

**Hypothesis** **3.**
*We expect indicator saprotroph and pyrophile species’ relative abundance to increase with decomposition (Figure 1).*


## 2. Materials and Methods

### 2.1. Site Description

Three managed upland (not wet) prairies on silty clay soil within 15 km of Eugene, Oregon were used (Table 1, Appendix A). The dominant species of grasses are shown in Table 1. Each prairie had a section that had a prescribed burn in the previous fall (2022), and a section that was unburned. For record keeping purposes, the different sections were given separate directional names (e.g., East, West), but the prairies with the same names were more or less contiguous, just burned in blocks. The controlled burns are used to control woody plant invasions and facilitate native prairie plant species [24]. The study sites were burned and unburned prairies on the East side of Mt. Pisgah and two locations within the Fern Ridge natural area: Eaton and Spire (see Appendix A). The prairies themselves were mostly composed of grasses and forbs (see Table 1 for specific species), with incursions of blackberry (*Rubus bicolor*) and shrubs/trees (*Quercus garryana, Crataegus* spp., *Fraxinus latifolia*).

### 2.2. Site Management Activities During the Experiment

Two of the prairies, Eaton and Spire, experienced mowing during the course of the experiment. Because mowing would destroy the litter bags, we picked them up the day before (11 September 2023) by placing each set of six into a separate new, dry paper bag. They were returned to the same locations the following day. Managers at the third prairie, Pisgah, hoped to burn it in August or September of 2023. Because conditions for controlled burns must be just right (low wind, sufficient crews, etc.), we were asked to remove our bags on 22 July 2023 and were unable to put them back out until 12 October 2023. Unfortunately, the conditions were never right to reburn this prairie.

### 2.3. Experimental Design

We used a full factorial design replicated at each of the three prairies (Pisgah, Eaton, and Spire, Eugene, OR, USA); within each of the three prairies, fire (yes, no) was fully crossed with litter treatment (autoclaved or not). The collected litter was sterilized by autoclaving at 121 °C for 15 min (liquid cycle). The 100 × 100 mm litter bags were made from nylon fabric, with a mesh size of <1 mm, and were closed with staples. Approximately 1.5–2 g of litter was placed into each bag before it was stapled shut. To account for the four staples used to close the bag after initial weighing, 0.13 g were added to each “before” weight, the average weight of the four staples based on five samples.

Litter was collected from each site, then put back at the site it was taken from to decompose. Approximately 50 g of litter was taken from both the burned and unburned section of each site, with the litter being taken from the vicinity of where it was to be set out again. Once the litter was bagged, the samples were mixed (homogenized) on a per site basis in the lab before half was autoclaved, so that there was similar material in each replicate. These litter bags were then placed under pre-existing litter and on the soil-surface in July, which is the time when the prairies’ above-ground biomass are dying back naturally. Samples were recovered at four different times post placement: after 3 months (5 October 2023), 6 months (4 February 2024), 9 months (5 May 2024), and 12 months (10–14 July 2024). Prior to weighing, green vegetation (new growth) was removed from the bags when necessary.

### 2.4. Nutrient Analyses

We measured carbon and nitrogen on a subset of the samples (from the three-month series) to determine if differences in decomposition were due to nutrient availability. Samples from Eaton and Spire were ground and homogenized, put to dry overnight in an oven at 105 degrees F, and then further ground by a bead milling machine (Fisherbrand Bead Mill 24 Homogenizer, Fischer Scientific Co., Pittsburgh, PA, USA). Seven milling balls were placed into each sample tube, and then run through the machine for two minutes at a speed of 3.40 m/s. Each tube was run through these settings twice. After bead milling, each sample was run through a FlashSmart Elemental Analyzer (Thermo Fisher Scientific, Fischer Scientific Co., Pittsburgh, PA, USA) to obtain the percentage composition of carbon and nitrogen for each sample.

### 2.5. Sequencing

Leaf litter samples collected at the 3- and 12-month periods were ground in UV-sterilized mortar and pestles along with liquid nitrogen. DNA was extracted from the 0.15 g litter samples using a DNeasy PowerSoil Pro kit (Qiagen, Germantown, MD, USA). The fungal specific primers ITS1F and ITS2 [25], modified with heterogeneity spacers and Illumina TruSeq barcode stubs [26], were used to amplify the ITS1 region in polymerase chain reactions (PCR). An initial round of PCR was performed in 20 uL reaction volumes with the GoTaq Green Master Mix (Promega Corp., Madison, WI, USA) at an initial denaturation step of 1 min at 94 °C, followed by 30 cycles of 45 s at 94 °C, 1 min at 54 °C, and 1 min 30 s at 72 °C, followed by a final extension of 72 °C for 10 min. A second round of PCR was performed for the attachment of unique multiplexed barcodes following the above protocol but modified for 12 cycles. After each round of PCR, the products were visualized with SafeView Classic (Applied Biological Materials Inc., Richmond, BC, Canada) on a 1% agarose gel via electrophoresis. The products were quantified using Quant-iT PicoGreen (ThermoFisher, Waltham, MA, USA) on a SpectraMax M5E Microplate Reader (Molecular Devices, San Jose, CA, USA) and pooled at equimolar concentrations. The pools were purified using Mag-Bind TotalPure NGS Magnetic Cleanup Beads (Omega Bio-tek, Norcross, GA, USA) at 0.85X for amplicon size selection. The pools were sequenced on the Illumina MiSeq at the University of Oregon’s Genomics and Cell Characterization Core Facility (Eugene, OR, USA). The sequencing data are available via The National Library of Medicine NCBI: PRJNA1267473.

### 2.6. Bioinformatics

The sequences were returned demultiplexed. We used the Cutadapt program to remove primers [27], dada2 to denoise [28], and QIIME 2.0 [29] to cluster the sequences to a 99% match of the dynamic UNITE trained classifier (colinbrislawn github, UNITE, 2024). Using the FungalTraits database, we matched fungal genera to trait assignments (e.g., saprotrophs, pathogens, mycorrhizae; [30]). To enable the subsetting of pyrophilous fungi for the analysis of next generation sequencing, we used the literature to create a comprehensive list of fire-associated (pyrophilous) fungi known from the USA (Appendix A). While all of the fungi in the list are fire-associated according to the references cited, the ecology of that association is often poorly understood, and we thus use the list to highlight areas lacking data. The list includes macrofungi known to increase fruiting after a fire (the “traditional” pyrophilous definition) but also microfungi, such as *Penicillium* and *Aureobasidium*, that sequencing studies are showing an increase in abundance of, post-fire (e.g., [7,31,32]). We thus used the list to highlight where information was lacking. For downstream analyses, we then subsetted fungal OTUs into primary function saprotrophs, including all saprotroph types (4684), pyrophilous fungi (76), and other fungi (8117).

### 2.7. Data Analyses

We calculated the proportion of litter decomposed by subtracting the final weight from the initial weight and dividing that by the initial weight to measure the amount decomposed at each time point. In addition, we calculated k (decomposition rate, [33]) by taking the log base e of the ratio of the final to initial weight, dividing by the number of months the sample was decomposing, and multiplying everything by −1.

To determine the effects of burning on decomposition, we ran a generalized linear model (glm) with the proportion of decomposed litter as a response variable, burn and sterilization as interacting effects, and site, fresh weight, as well as the time the bags were in the field (months) as fixed effects. At the Pisgah site, the bags were removed soon after initial placement to allow for controlled burning, then replaced after the first collection period. Due to this major site difference, we included this site as a fixed effect instead of random. We subsetted each sample collection time and reran the model without months left out, to measure whether these effects contributed to differences in decomposition across each time period. Finally, to measure whether nutrient differences were contributing to differences in composition, we ran a glm with the ratio of C:N as responses to the decomposition weight, burn, and site. Analyses and graphing were performed with R (version 4.2.2), using the following packages: ggplot, tidyverse, ggpubr, rstatix, emmeans.

For all fungal subsets, we calculated the Shannon–Weiner diversity using the vegan package in R [34] and ran a glm to measure the responses to site, date of removal, autoclaving, and burn treatments, with an interaction between burn and autoclave treatments. We calculated the robust Aitchison distance for saprotroph- and pyrophile-subsetted ASV’s [35], then ran permanovas using adonis2 [34] to measure the community responses to burn, autoclaving, burn*autoclaving, and site. To identify the ASV’s that had significant variations explained by burn and autoclave treatments, we used multipatt in the R package indicspecies R 4.4.2 [36]. We measured the contribution of the saprotroph and pyrophilous fungal diversity to the decomposition rate with a simple generalized linear model (glm), with 1 added to the log transformed diversity measurement to shift the residuals closer to normal.

## 3. Results

### 3.1. Does Litter Decomposition Depend on Treatment? Hypothesis 1

Litter decomposition rate was not significantly affected by fire (Table 2). However, the decomposition rate was significantly lower in autoclaved litter, compared with untreated litter (*p* << 0.001, Figure 2). Decomposition rates were not constant (months out, *p* << 0.001); they were faster at the beginning (Figure 2). The ratio of C:N was not significantly affected by burn treatments in the burned sites, though it decreased as decomposition rates increased (*p* = 0.01), with significant variations among the sites (*p* = 0.01; Appendix A).

### 3.2. Does Fungal Diversity Depend on Treatment, and if So, Does Fungal Diversity Contribute to Decomposition? Hypothesis 2–3

The relationships between fungal diversity and treatments depended on the particular functional group. There was no significant main effect of fire for either pyrophilous or saprotrophic fungi, but there was a significant main effect of autoclaving; it decreased the diversity of saprotrophs, but increased the diversity of pyrophiles (Table 3, Figure 3). Importantly, there was a significant interaction between fire and autoclaving for both functional groups (Table 3). In burn, the saprotroph diversity was significantly lower (Figure 3A) when autoclaved, whereas pyrophilous diversity was significantly higher (Figure 3B) when autoclaved.

### 3.3. How Does Fire Alter Saprotrophic and Pyrophilous Fungal Community Composition in Prairies?

The ten most abundant saprotroph OTUs were a *Cadophora* sp., *Davidiellomyces* sp., *Sclerostagonospora* sp., or *Conioscypha* sp., with several OTUs matched only to the genus *Sclerostagonospora*. The ten most abundant pyrophilous OTUs were all identified as the species *Aureobasidium pullulans*. We note that duplicate OTU identification is likely due to 99% taxonomy clustering methods. In the permanova, the saprotrophic fungal composition did not differ significantly between burn and autoclave treatments, or sites (Appendix A). However, the pyrophilous fungal composition was significantly different between autoclave treatments (*p* = 0.01), with significant effects of site (*p* = 0.001). Saprotroph diversity did not correlate significantly with the decomposition rates (*p* = 0.69, Figure 4A), but pyrophilous fungal diversity had a significant positive correlation (*p* << 0.001, Figure 4D).

In the indicator species analysis, we found a higher relative abundance of a species in the saprotrophic genus *Sclerostagonospora* (only a 95.45% match to the type of *Sclerostagonospora fusiformis*) in the unburned sites, compared with burned sites (*p* = 0.047; Figure 4B) and a higher relative abundance of a different *Sclerostagonospora* ASV in untreated litter, compared with autoclaved litter (*p* = 0.015; Figure 4C). In addition, two ASV’s, identified as *Aureobasidium pullulans,* had higher relative abundances in the burned sites, compared with the unburned sites (*p* = 0.05, *p* = 0.027; Figure 4E); eight ASV’s identified as *Aureobasidium pullulans* also had higher relative abundances for autoclave-treated litter, compared with untreated litter (*p* << 0.001; Figure 4F).

## 4. Discussion

Our results point to the importance of pyrophilous fungi in decomposition. Our first hypothesis was that litter in a burned prairie would have lower decomposition, due to both decreased available plant biomass post-fire and a decreased number of decomposer species [37]. However, we found no significant differences in either saprotrophic or pyrophilous fungal diversity between burn treatments alone (Table 3). For hypothesis two, saprotrophic diversity was lower in the autoclaved samples at burned sites, with the opposite response for pyrophiles (Figure 3). Hypothesis three was supported for pyrophilous fungi, as their diversity increased along with increased decomposition rates (Figure 4D). These effects are likely the result of differences in fungal community members, such as the *Sclerostagonospora* spp. and *Aureobasidium pullulans* indicator species, which had significant differences in their relative abundance between burned and unburned sites (Figure 4B,E), as well as for autoclaved litter (Figure 4C,F).

Our attempt to reduce starting fungal diversity, the autoclaving treatment, had unexpected effects on the pyrophilous fungal contribution to decomposition. While total litter saprotroph diversity was significantly lower in the autoclaved samples at burned sites (Figure 4A), pyrophilous fungal diversity was higher (Figure 4B). This suggests that either the pyrophilous fungi were not all killed by autoclaving as has been found in some studies (e.g., [19,38]), or that they were better able to break down the high heat-treated compounds in the autoclaved litter and thus rapidly colonized it [7]. Interestingly, the most common pyrophilous fungus we found, *Aureobasidium pullulans*, was found in ash at Mt. St. Helens shortly after it exploded [39], but again we do not know if this was due to heat tolerance or rapid invasion post-fire. This species is often found as an endophyte in plant tissue, offering powerful anti-pathogen properties to its living hosts [40], suggesting it may play an important role in plant recovery post-fire. The implications of our findings are threefold: firstly, future studies should consider how particular fungal groups might respond to sterilization methods; secondly, there are likely ecological implications of some pyrophilous fungi surviving such sustained high heat [40] and lastly, the term phoenicoid fungi may be more appropriate than pyrophilous as was argued by Carpenter et al. [41].

Do these fungi require heat or fire? Traits of fire- and heat-associated fungi have recently been gaining more attention (e.g., [6,7,42]), and there is a lot to learn about them, as indicated by the number of question marks in our literature review (Appendix A). In addition to fire-association, we included traits such as whether the fungus is known to be an endophyte, whether it is known to metabolize burnt organic matter, and whether it occurs early (days to a few months) or later in succession to fires. While we built the table to enable the subsetting of pyrophilous fungi in next generation sequencing studies, it can also be used to guide future research into the traits of these fascinating fungi. Our review (Appendix A) also shows a bias towards studying macrofungi that fruit after a fire because they are easy to see. Future studies comparing paired burned and unburned sites could facilitate the discovery of more fire-associated microfungi if they listed the most common taxa in accessible tables.

Decomposition rates declined more rapidly at burned sites. Initially, litter decomposition rates were significantly higher in untreated litter at burned sites, though the difference between autoclave treatments disappeared over time (Figure 2). In an attempt to explain this relationship, nutrient analyses were performed on the litter bags from the initial removal to see if nutrient availability could account for this difference. In particular, we thought autoclaving had the potential to alter nutrient availability in litter samples. However, we found no differences in the C:N ratio between burned and unburned litter samples collected three months after the fire (Appendix A). Other studies have found a relationship between fire, decomposition, and nutrient availability (e.g., [43,44,45]), but they had sampled immediately after the fire, whereas we sampled 9 months post-fire. More interestingly, the diversity of pyrophilous fungi, as well as the relative abundance of *A. pullulans*, was higher in autoclaved samples at burned sites. These findings were positively correlated (Figure 4D), suggesting that pyrophilous fungi were significant contributors to decomposition when litter had undergone high heat. While differences in decomposition between burned and unburned sites in those studies were a result of the fire affecting nutrient availability and ratios, our data indicate that that was not the case here. Our findings indicate that the differences in decomposition rates were explained by pyrophilous fungi, rather than nutrients.

Autoclaved samples generally had lower rates of decomposition than the untreated samples (Figure 3), supporting our hypothesis that fungal endophytes inside the plant cells that remained alive would initiate the decomposition process (Hypothesis 1). However, autoclaving does more than killing endophytic fungi, it also changes chemical and tissue compositions via the breakdown of lignin. In fact, there are many studies that have examined the effect of autoclaves on cellular composition, many of which have found that the process of sterilization by autoclave can cause tissues to break down [18,46,47]. In future studies, we suggest a methodology to measure the various relative effects of autoclaving. Interestingly, we found a greater diversity of pyrophilous fungi in our autoclaved litter from the burned sites. This suggests that the colonization of pyrophilous fungi was enhanced in the post-fire conditions at the burned sites, and this is consistent with their ability to break down heat-altered plant compounds [5,7]. It is surprising to have detected more interactions between autoclave and burn treatments, rather than the predicted general lower diversity in autoclaved samples. This may partly be a sample size effect resulting from the ⅓ fewer samples from the three-month collections having a weaker response, whereas saprotroph diversity was significantly lower in the autoclaved litter and burned sites from the 12-month sampling, which had all the samples (see Appendix A). Finally, studies investigating the success rates of autoclaving as a means of sterilization highlight the many ways in which autoclaves could malfunction, thus resulting in surviving microorganisms [48].

## 5. Conclusions

Prescribed burns are a key part of maintaining native PNW prairies; fire keeps trees and shrubs from turning into forests, and fire also encourages growth of some native species, such as camus that have evolved under thousands of years of native American prairie burning practices [24,49]. Outside of prescribed, intentional burns, wildfires are expected to increase under anthropogenic climate change conditions [50], and generally, fires are predicted to become larger, more intense, and happen with greater frequency [51]. With the understanding that fire is something these prairies will be regularly exposed to, whether intentional or not, it is important to understand the effects of fire across different taxa and not just plants. Having insights into how decomposition might change in areas that have been burned can give the managers of these landscapes tools to be ready for the larger scale implications of these changes in decomposition. For example, our data suggest that controlled burning does not significantly affect decomposition rates, as the heat triggers pyrophilous fungi to contribute to decomposition when more fire-sensitive saprotrophs cannot. These findings support the use of intentional burning as a maintenance strategy in Oregon prairies.

## Figures and Tables

**Figure 1 microorganisms-13-01834-f001:**
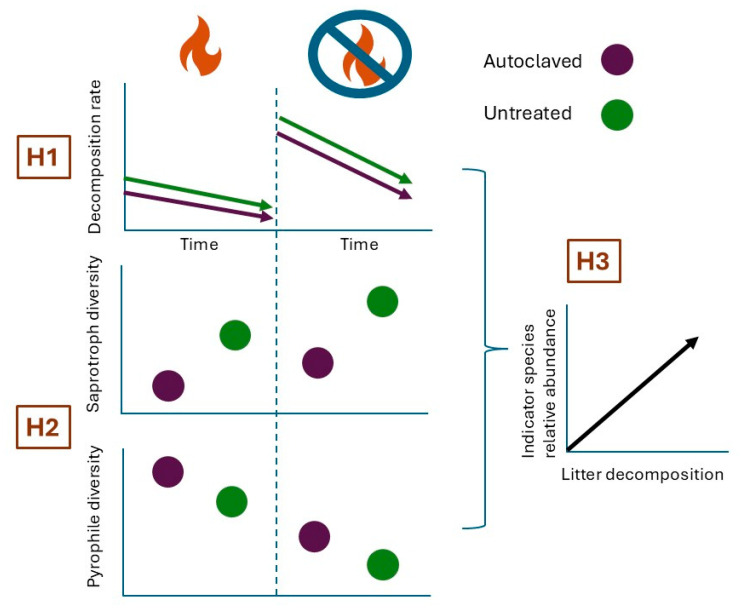
Conceptual diagram of hypotheses.

**Figure 2 microorganisms-13-01834-f002:**
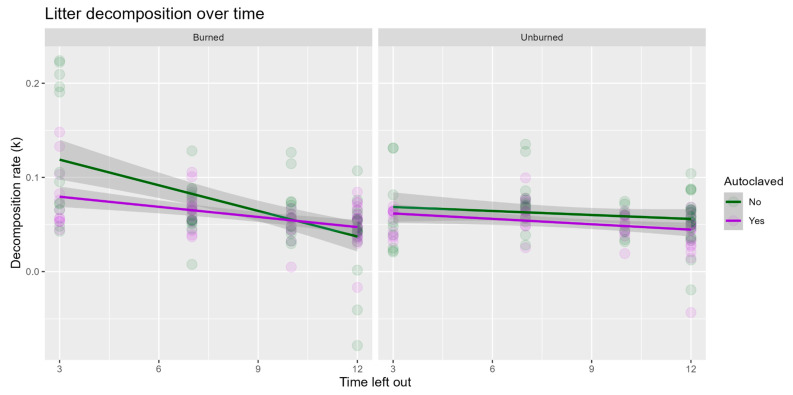
Litter decomposition rates over time. Litter decomposition rates generally declined over time (*p* << 0.001) and were significantly lower in autoclave-treated litter (*p* << 0.001). In the burned plots, decomposition rates were higher in the early sample collections (months out, *p* << 0.001), with no significant interaction between burn and autoclave treatments.

**Figure 3 microorganisms-13-01834-f003:**
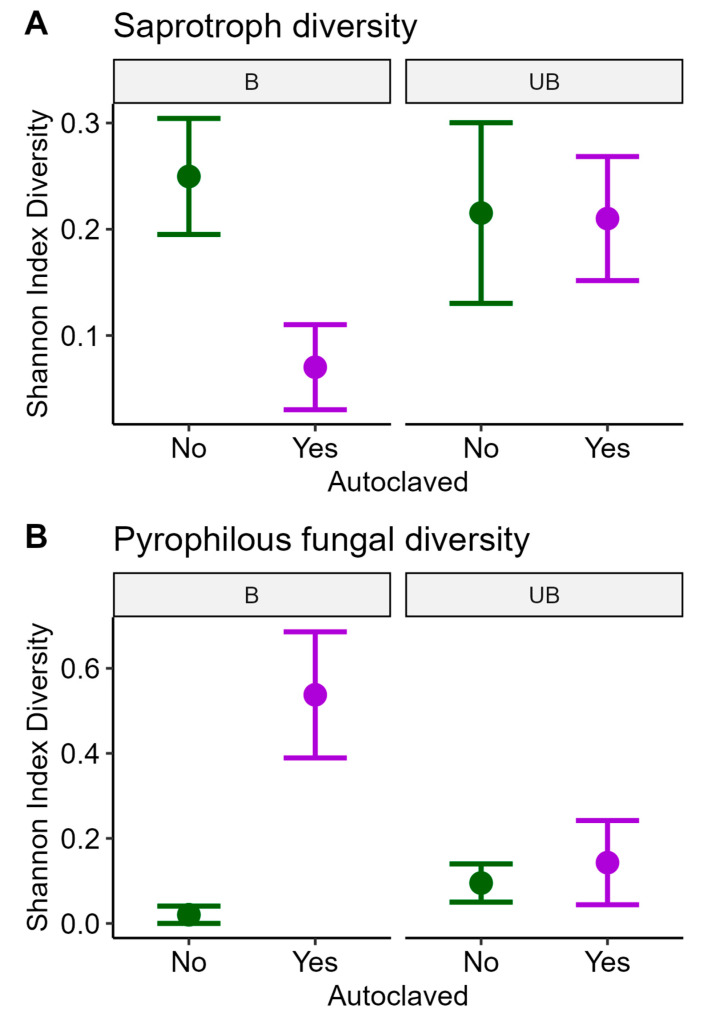
Saprotroph and pyrophilous fungal diversity response to the interaction of autoclave and burn treatments. Saprotroph diversity (**A**) was significantly lower in autoclaved samples (*p* = 0.007), with an interaction between burn and autoclave treatments (*p* = 0.035) and marginal variation among sites (*p* = 0.066). Pyrophilous fungal diversity (**B**) was significantly higher in autoclaved samples (*p* << 0.001), with a significant interaction between burn and autoclave treatments (*p* = 0.024). B = Burned, UB = Unburned. Purple indicates autoclaved samples, green indicates untreated.

**Figure 4 microorganisms-13-01834-f004:**
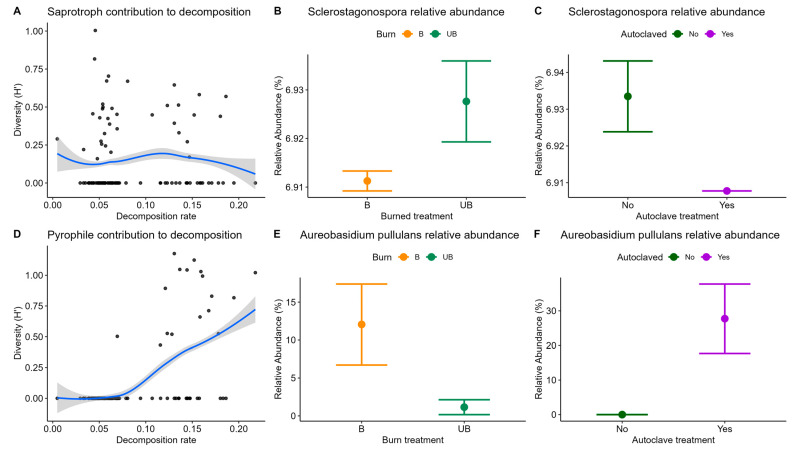
Fungal saprotroph and pyrophile relationships to decomposition rates. Saprotroph diversity had no relationship with decomposition rate ((**A**), *p* = 0.69). *Sclerostagonospora* spp. relative abundances were significantly higher in unburned ((**B**), *p* = 0.05) and untreated litter samples ((**C**), *p* = 0.015). Pyrophilous fungal diversity significantly increased with decomposition rates ((**D**), *p* << 0.001). *Aureobasidium pullulans* relative abundances were significantly higher in burned ((**E**), *p* = 0.05) and autoclaved litter ((**F**), *p* = 0.009). Many samples did not have identified saprotrophs or pyrophiles (zeros in (**A**,**D**)).

**Table 1 microorganisms-13-01834-t001:** Localities. Information about each replicate site. Pictures of each are in Appendix A. Common plant species from observations taken at each site at the time of litter collection. A = Absent, I = Invasive, F = Forb, G = Grass, N = Native, P = Present.

Site	Name	Burned?	*Aira caryophyllaea* (I, G)	*Arrhenatherum elatius* (I, G)	*Anthoxanthum odoratum* (I, G)	*Daucus carota* (I, F)	*Eriophyllum lanatum* (N, F)	*Festuca roemeri* (N, G)	*Hypochaeris radiata* (I, F)	*Vulpia* spp. (I, G)	Lat	Long	Elevation (m)
N	Pisgah	No	P	A	A	P	A	P	P	P	43.999809	122.943102	161
South	Pisgah	Yes	P	A	A	P	A	P	P	P	43.996911	122.944631	161
South	Eaton	No	P	A	A	P	P	P	P	P	44.101167	123.259355	113
North	Eaton	Yes	P	P	A	P	A	P	P	P	44.102437	123.259369	113
East	Spire	No	P	P	P	P	A	P	P	P	44.100123	123.262949	113
West	Spire	Yes	P	A	A	P	P	P	P	P	44.098619	123.264431	113

**Table 2 microorganisms-13-01834-t002:** Linear model, litter decomposition rate response to burn and autoclave treatments.

Decomposition Rates				
	Estimate	Std Error	t Value	Pr (>F)
Burn	−0.007	0.006	−1.294	0.20
Autoclaved	−0.010	0.005	−1.757	8.02 × 10^−2^
Months.out	−0.004	0.001	−5.691	3.68 × 10^−8^
Fresh.Weight	0.006	0.007	0.825	0.41
Site-Pisgah	−0.011	0.005	−2.092	0.04
Site-Spire	−0.010	0.005	−1.883	0.06
Burn: Autoclaved	−0.001	0.008	−0.081	0.94

**Table 3 microorganisms-13-01834-t003:** Saprotroph and Pyrophilous Fungal diversity responses.

Diversity Response								
Saprotrophs					Pyrophiles			
	Estimate	Std. Error	t value	Pr (>|t|)	Estimate	Std. Error	t value	Pr (>|t|)
Burn	−0.056	0.060	−0.935	0.352	0.088	0.058	1.516	0.133
Autoclaved	−0.156	0.057	−2.736	0.007	0.286	0.056	5.147	1.23 × 10^−6^
Time	−0.079	0.056	−1.403	0.163	−0.402	0.055	−7.317	5.10 × 10^−11^
Site	0.106	0.057	1.856	0.066	−0.025	0.056	−0.449	0.654
Burn: Autoclaved	0.187	0.088	2.130	0.035	−0.209	0.086	−2.445	0.016
Residual dev.		5.44, 106 df				5.15, 106 df		

## Data Availability

The data presented in this study are openly available in [NCBI] at [https://www.ncbi.nlm.nih.gov/bioproject/PRJNA1267473/ accessed on 24 May 2025], reference number [PRJNA1267473].

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
