# Peer review of "Litter Decomposition in Pacific Northwest Prairies Depends on Fire, with Differential Responses of Saprotrophic and Pyrophilous Fungi"

_microorganisms, 2025, doi:10.3390/microorganisms13081834_

Round 1

Reviewer 1 Report

Comments and Suggestions for Authors

method:

Lines 112-114: Sampling interruption at Pisgah site due to fire management (Section 2.2), although included as a fixed effect in the model, the impact of this bias on the results still needs to be explained in the discussion of limitations. Is the time effect of decomposition rate amplified as a result?.

The list of fire loving fungi (Table A1) includes species with unclear fire associations (such as Aureobasidium pullulans), and the screening criteria need to be specified in the method.

Lines 169-170: Supplementary fungal trait data source: The version and update date of the FungalTraits database [29] should be indicated.

Line 182: It is necessary to supplement the residual distribution test results of the GLM model to ensure compliance with the normality hypothesis.

result

Figure 3: Clearly labeled "B=incinerated, UB=not incinerated". The current legend is missing.

Figure 4: Integrating subgraphs (such as merging B/C and E/F) to avoid information fragmentation; Supplement coordinate axis units. Such as' Relative abundance (%) '.

Why is the table presented in the form of images? Between lines 236-238, there is a presence of 3.4 between the text and image. And all attachments should be placed after the references. The issue of charts requires the author's attention.

Discussion:

Lines 327-331: Emphasize that it may alter the chemical properties of litter (such as lignin degradation) rather than just sterilization (cited [18,44-45]), and analyze whether this explains the higher diversity of fire loving fungi in high-pressure sterilization samples.

Lines 338-339: Since you mentioned it, it is necessary to further discuss why diversity of fire loving fungi (rather than saprophytic fungi) is positively correlated with decomposition rate.

Writing:

Unified use of 'pyrophilous fungi' (not 'phoenicoid fungi').

reference:

All references are incorrect, please carefully revise them using the journal template.

Author Response

Comments 1: method:

Lines 112-114: Sampling interruption at Pisgah site due to fire management (Section 2.2), although included as a fixed effect in the model, the impact of this bias on the results still needs to be explained in the discussion of limitations. Is the time effect of decomposition rate amplified as a result?.

Response 1: We agree that Pisgah’s differences likely impact the results. For this reason we included the time each sample was out at the field sites in the model – for Pisgah, those were 3, 7, 10, and 12 months. Then, for fungal response analyses we only use samples from 3 and 12 months, which represent the same amount of time left out across sites.

To address the question of time effect of decomposition rate, we include a figure and linear model output for the same analysis with Pisgah removed. This is now in the supplementary information, as Fig. A5 and Table A4.

Comments 2: The list of fire loving fungi (Table A1) includes species with unclear fire associations (such as Aureobasidium pullulans), and the screening criteria need to be specified in the method.

Response 2: We have added the following sentences to the methods: “While all of the fungi in the list are fire associated according to the references cited, the ecology of that association is often poorly understood, and we thus use the list to highlight areas lacking data. The list includes macrofungi known to increase fruiting after fire (the “traditional” pyrophilous definition) but also microfungi, such as Penicillium and Aureobasidium, that sequencing studies are showing increase in abundance postfire (e.g., Enright et al. 2022; Barbour et al 2023; Packard et al. 2023). 

Note for the reviewer, the Barbour and Packard references show massive increases in abundance of Aureobasidium yeasts postfire. These yeasts are common in sequencing data.

Comments 3: Lines 169-170: Supplementary fungal trait data source: The version and update date of the FungalTraits database [29] should be indicated.

Response 3: The version and update date of the FungalTraits database is inherent to the cited publication. We used the version in the cited 2020 paper by Põlme et al.

Comments 4: Line 182: It is necessary to supplement the residual distribution test results of the GLM model to ensure compliance with the normality hypothesis.

Response 4: We now include histograms in the appendix (Fig. A6) for decomposition rate response variable, as well as saprotroph and pyrophilous fungal diversity distributions with the transformation described on lines 208-209. We tried many different transformation types for diversity metrics, and found the log(diversity + 1) to be the closest to normal distribution.

result

Comments 5: Figure 3: Clearly labeled "B=incinerated, UB=not incinerated". The current legend is missing.

Response 5: We have edited the figure caption to specify that B=burned and UB=unburned.

Comments 6: Figure 4: Integrating subgraphs (such as merging B/C and E/F) to avoid information fragmentation; Supplement coordinate axis units. Such as' Relative abundance (%) '.

Response 6: We have edited the y-axis titles for B/C and E/F to be consistent. However, we politely decline the suggestion to merge those graphs, as their current orientation makes it easiest to refer to them in the text.

Comments 7: Why is the table presented in the form of images? Between lines 236-238, there is a presence of 3.4 between the text and image. And all attachments should be placed after the references. The issue of charts requires the author's attention.

Response 7: We included the tables in the form of .jpg images due to the formatting discrepancies that can occur when transferring excel data tables to word documents. We have added tables as formal tables into the word document now.

All main text tables and figures have now been placed following their first reference in the text itself. We apologize for any inconvenience the previous ordering caused.

Discussion:

Comments 8: Lines 327-331: Emphasize that it may alter the chemical properties of litter (such as lignin degradation) rather than just sterilization (cited [18,44-45]), and analyze whether this explains the higher diversity of fire loving fungi in high-pressure sterilization samples.

Response 8: We edited this area to discuss the alterations autoclaving can cause, which we did not measure in this study.

Comments 9: Lines 338-339: Since you mentioned it, it is necessary to further discuss why diversity of fire loving fungi (rather than saprophytic fungi) is positively correlated with decomposition rate.

Response 9: We explain this in L329-330.

Writing:

Comments 10: Unified use of 'pyrophilous fungi' (not 'phoenicoid fungi').

Response 10: We are not sure what the reviewer is suggesting. We have used ‘pyrophilous’ fungi throughout the text, only mentioning ‘phoenicoid’ as an alternative word for these fungi in the discussion. Some references recommend phoenicoid instead of pyrophilous because extreme heat, not fire is the mechanism that causes some of them to increase as is the case with our autoclaving and also volcanic eruptions (e.g., Carpenter, S. E., J. M. Trappe, and J. Ammirati. 1987. Observations of fungal succession in the Mount St. Helens devastation zone, 1980-1983. Canadian Journal of Botany-Revue Canadienne De Botanique 65:716–728.

reference:

Comments 11: All references are incorrect, please carefully revise them using the journal template.

Response 11: We used the Zotero Microorganisms template to correct the references.

Reviewer 2 Report

Comments and Suggestions for Authors

This manuscript addresses an interesting problem and should be published in Microorganisms. The study was conducted on three prairies in the USA. Each prairie had a section with controlled burns and a section that was unburned. Litter for study was collected from both sections, which in one of the variants was autoclaved. Litter decomposition was examined, nutrient analyses were performed, and fungal OTUs were determined (by sequencing). The data were well analyzed using statistical tools. The results were presented in detail and clearly. The effect of treatments on litter decomposition rate and fungal diverity was presented. Saprotrophs and pyrophilous fungi were considered separately. The list of fungi with different life styles presented in Table A1 based on the literature is very valuable. However, I would like to point out that the Authors formulated three hypotheses (lines 86-92), but they do not clearly address all of them in the Results or Discussion section. This should be supplemented. In addition, the Discussion section should indicate for several pyrophilous fungi what specific strategies enable them to increase in frequency after fire. For example, for Aureobasidium pullulans it may be its frequent occurrence as an endophyte inside plant tissue. There are typographical errors in several places in the text (see Remarks).

Remarks

ine 92 composition. (Fig. 1)

Line 104 Rubus bicolor, Quercus garryana, Crataegus , Fraxinus latifolia – it should be italic

Line 169 it should be saprotroph, instead of saprotoph

Line 169 should be used singular or plural for uniformity (saprotroph, pathogens …)

Line 223 Cadophora Davidiellomyces Sclerostagonospora Conioscypha - it should be italic, this applies to fungi throughout the manuscript

Line 223-224 The most abundant saprotroph OTUs … The most abundant pyrophilous OTUs were all identified as Aureobasidium pullulans. This text in its current form is unclear, it requires clarification.

Line 237 3.4. Figures, Tables and Schemes the title of this section should be above Figure 1 (in the current version of Figure 1 there is no caption)

Line 239 Table 1  spp. – it should be not italic

Line 256 Figure 3 – B, UB  should be explained

Line 279 spp. – not italic

Line 315 fire. (Fig. A3).

Line 326 it is difficult to understand which hypothesis contained such text (see hypotheses 1-3 in lines 86-92)

Table A1

- synonym for Morchella eximia requires supplementation

-aff. – should be not italic

Author Response

Comments 1: This manuscript addresses an interesting problem and should be published in Microorganisms. The study was conducted on three prairies in the USA. Each prairie had a section with controlled burns and a section that was unburned. Litter for study was collected from both sections, which in one of the variants was autoclaved. Litter decomposition was examined, nutrient analyses were performed, and fungal OTUs were determined (by sequencing). The data were well analyzed using statistical tools. The results were presented in detail and clearly. The effect of treatments on litter decomposition rate and fungal diverity was presented. Saprotrophs and pyrophilous fungi were considered separately. The list of fungi with different life styles presented in Table A1 based on the literature is very valuable. However, I would like to point out that the Authors formulated three hypotheses (lines 86-92), but they do not clearly address all of them in the Results or Discussion section. This should be supplemented. In addition, the Discussion section should indicate for several pyrophilous fungi what specific strategies enable them to increase in frequency after fire. For example, for Aureobasidium pullulans it may be its frequent occurrence as an endophyte inside plant tissue. There are typographical errors in several places in the text (see Remarks).

Response 1: We appreciate the reviewer’s recognition of the value of our paper. We agree that the results and discussion could more clearly address the hypotheses, and we made some changes to tie these sections together better. The consideration of A. pullulans lifestyle is an interesting point, and we happily add discussion about this in the text (L300).

Remarks

Comments 2: Line 92 composition. (Fig. 1)

Response 2: We aren’t sure what the reviewer is saying about this line. We note that the word ‘decomposition’ begins on line 91 and continues down on line 92.

Comments 3: Line 104 Rubus bicolor, Quercus garryana, Crataegus , Fraxinus latifolia – it should be italic

Response 3: Thank you for catching this. The italicized words must have undone when transferring the text into the template document. We have gone through and italicized all genera and species names.

Comments 4: Line 169 it should be saprotroph, instead of saprotoph

Response 4: Thank you for bringing this typo to our awareness. It has been properly changed to ‘saprotroph.’

Comments 5: Line 169 should be used singular or plural for uniformity (saprotroph, pathogens …)

Response 5: Amend to above response, we have changed it to read “(e.g. saprotrophs, pathogens, …)”

Comments 6: Line 223 Cadophora Davidiellomyces Sclerostagonospora Conioscypha - it should be italic, this applies to fungi throughout the manuscript

Response 6: We have changed all genus and species names to italics.

Comments 7: Line 223-224 The most abundant saprotroph OTUs … The most abundant pyrophilous OTUs were all identified as Aureobasidium pullulans. This text in its current form is unclear, it requires clarification.

Response 7: This is now on lines 253-255. We clarify here that any overlapping OTU taxonomy assignments are likely due to the 99% clustering threshold we use and the fact that short reads (<300bp) are generated by next generation sequencing (here Illumina). This is common in sequencing data, where the closest known match to DNA sequence is the same species for multiple OTUs.

Comments 8: Line 237 3.4. Figures, Tables and Schemes the title of this section should be above Figure 1 (in the current version of Figure 1 there is no caption)

Response 8: Figures and tables are now immediately following the section where they are first mentioned.

Comments 9: Line 239 Table 1  spp. – it should be not italic

Response 9: We have corrected the error.

Comments 10: Line 256 Figure 3 – B, UB  should be explained

Response 10: We have added the meaning of B and UB in the figure description.

Comments 11: Line 279 spp. – not italic

Response 11: The “spp.” in this line has been un-italicized.

Comments 12: Line 315 fire. (Fig. A3).

Response 12: We removed the period after fire.

Comments 13: Line 326 it is difficult to understand which hypothesis contained such text (see hypotheses 1-3 in lines 86-92)

Response 13: This is referring to hypothesis 1. We have made some additions to the text to follow the hypothesis structure we outline in the beginning.

Comments 14: Table A1

- synonym for Morchella eximia requires supplementation.

Response 14: The requested information was already in the table, in the references cited for that line.  See reference 64=Richard et al. 2015, Page 364. 

Comments 15: -aff. – should be not italic

Response 15: The “aff.” in the table has been un-italicized.

Reviewer 3 Report

Comments and Suggestions for Authors

The paper explores how fire and litter decomposition affect fungal communities , particularly focusing on saprotrophic and pyrophilous (fire-associated) fungi , in Pacific Northwest prairies . The study includes experimental manipulations such as burning plots and autoclaving litter bags to assess microbial community dynamics and decomposition rates over time.

My comments:

Please adjust your presentation style for tables to meet the journal's standards. I suggest looking at other papers in the Microorganisms journal for reference.

The carbon and nitrogen are measured, but the data are not clearly shown or discussed in the paper. There is no mention of EC, pH, or temperature, as these factors heavily influence microbial populations.

Please add more details on soil type, vegetation cover, or moisture levels across sites, which would strengthen the environmental context.

While the statistical methods are sound, some results lack full reporting of test statistics (e.g., F-values, degrees of freedom). Consider adding confidence intervals or effect sizes where applicable.

More explicit discussion of mechanisms (e.g., heat-stimulated spore germination, charcoal metabolism) would enhance the biological interpretation.

Potential impacts of autoclaving beyond sterilization, such as chemical changes in litter, should be mentioned in the limitations.

The degree symbol needs to be fixed throughout the manuscript.

Line 167. QIIME 2.0 needs a reference; please add it.

Line 207: Why are there two "<<" signs for the p value?

Author Response

Comments 1: The paper explores how fire and litter decomposition affect fungal communities , particularly focusing on saprotrophic and pyrophilous (fire-associated) fungi , in Pacific Northwest prairies . The study includes experimental manipulations such as burning plots and autoclaving litter bags to assess microbial community dynamics and decomposition rates over time.

My comments: Please adjust your presentation style for tables to meet the journal's standards. I suggest looking at other papers in the Microorganisms journal for reference.

Response 1: We have changed all tables to be formatted within the Microorganisms guidelines.

Comments 2: The carbon and nitrogen are measured, but the data are not clearly shown or discussed in the paper. There is no mention of EC, pH, or temperature, as these factors heavily influence microbial populations.

Response 2: We report C and N models and a figure on their effects on decomposition rate in the appendix (Table A2, Figure A3). We did not measure EC, pH, or temperature, unfortunately.

Comments 3: Please add more details on soil type, vegetation cover, or moisture levels across sites, which would strengthen the environmental context.

Response 3: Dominant grasses are listed in Table 1 along with elevation and location.  We added to the text that all the prairies are within 15 km of Eugene, of upland (not wet type) and on silty clays.  It can be inferred by location that they have the same climate and weather.

Comments 4: While the statistical methods are sound, some results lack full reporting of test statistics (e.g., F-values, degrees of freedom). Consider adding confidence intervals or effect sizes where applicable.

Response 4: We include standard error estimates in the statistical output tables.

Comments 5: More explicit discussion of mechanisms (e.g., heat-stimulated spore germination, charcoal metabolism) would enhance the biological interpretation.

Response 5: We aren’t sure what the reviewer is suggesting to enhance the interpretation of. We have added to the discussion of the pyrophilous fungi in the discussion.

Comments 6: Potential impacts of autoclaving beyond sterilization, such as chemical changes in litter, should be mentioned in the limitations.

Response 6: We have added consideration on lines 342-343.

Comments 7: The degree symbol needs to be fixed throughout the manuscript.

Response 7: On our end the degree symbol looks to be correct.

Comments 8: Line 167. QIIME 2.0 needs a reference; please add it.

Response 8: Thank you for catching this mistake. We have added the reference for QIIME 2.0.

Comments 9: Line 207: Why are there two "<<" signs for the p value?

Response 9: Two “<<” signs indicates very much less than 0.001. For example some of these numbers are E-5 or 6.

Round 2

Reviewer 1 Report

Comments and Suggestions for Authors

I agree to publish the revised content.

Reviewer 3 Report

Comments and Suggestions for Authors

Thank you for implementing the comments.